# Effects of Mutual Coupling on Gain and Beam Width of a Linear Array of a Dielectric Resonator Antenna for Main Beam Scanning Applications

**DOI:** 10.3390/s22186820

**Published:** 2022-09-09

**Authors:** Jamal Nasir, Aftab Ahmad Khan, Shoaib Khaliq, Muhammad Bilal Qureshi, Irfan Ullah, Leo Liu, Raheel Nawaz

**Affiliations:** 1Department of Electrical and Computer Engineering, COMSATS University Islamabad, Abbottabad Campus, Abbottabad 22010, Pakistan; 2Business and Law, Manchester Metropolitan University, Manchester M15 6GH, UK; 3Staffordshire University, Stoke-on-Trent ST4 2DE, UK

**Keywords:** half-power beam width, phase array, rectangular dielectric resonator antenna array

## Abstract

The effects of mutual coupling on the scanning characteristics of a four-element linear rectangular dielectric resonator antenna array (RDRA) are investigated for different inter-element spacing in this work. In particular, the gain and half-power beam width (HPBW) of an RDRA are studied for various scan angles in the E- and H-plane configurations. It is shown that for both the E and H planes, mutual coupling has an adverse effect on the performance of both phased array configurations. The H-plane array, however, is more stable than the E-plane array in terms of a gain and beam width performance comparison. The HPBW increases and gain decreases more in the E plane than the H plane when the scan angle is increased.

## 1. Introduction

In the last couple of years, dielectric resonator antennas (DRAs) have attracted considerable attention because of the availability of high-tech cutting and fabrication technologies. In addition, a lot of research has been conducted on DRAs covering different aspects, such as MIMO DRAs [1,2], circularly polarized DRAs [3,4], multi-band [5] and UWB DRAs [6,7], to name a few. The state of the art developments are covered in [8,9]. A DRA as an isolated radiator has a maximum gain of 5–6 dBi; because of this, DRAs in different array configurations have been studied for increased gain applications [10,11,12,13,14]. The performance of an antenna array, in terms of directivity, bandwidth, frequency of operation, beam width, gain and efficiency, is severely affected by the electromagnetic interaction (mutual coupling) between the array elements. Being 3D structures, DRAs have a higher mutual coupling behavior than printed antennas [15,16]. The simulation and experimental studies of mutual coupling effects for two-element linear DRA arrays in various configurations are carried out in [17,18]. The broadside radiation pattern and gain analysis for four-element linear DRA arrays using different feed arrangements (slot, probe, microstrip) are investigated in [19]. However, the effects of mutual coupling on gain and half-power beam width (HPBW) for main beam scanning applications have not been studied.

Thus, the objective of this work is to investigate the effects of mutual coupling on the gain and HPBW of main beam scanning for linear rectangular DRA (RDRA) arrays in the E- and H-plane configurations. Moreover, the main lobe scanning effects on the beam forming pattern of 1 × 4 linear RDRA will be explored.

## 2. Main Beam Scanning of a 1 × 4 Linear RDRA

The main contribution of this research work is to explore the mutual coupling effects on the gain and HPBW of four-element linear RDRA arrays for various scan angles of the main beam. The inter-element spacing in the E and H planes is chosen to be 0.5 λ and 0.32 λ, where λ is the free space wavelength. These inter-element spacings correspond to a low and a very high mutual coupling environment. To explore these effects, the four-element linear RDRA array shown in Figure 1 will be considered. The inter-element spacing between the array elements is 0.5 λ and 0.32 λ; and θs is the peak direction of the main beam from the z-axis.

An RDRA with dimensions a = 25.5 mm, b = 20 mm, height = 22 mm and εr = 15 is fed by a probe feed whose height is 11.5 mm. The RDRA is placed on the top of FR4 substrate with εr = 4.6 and a height of 1.6 mm, as shown in Figure 1. The dimension of the substrate is 40 (L) × 40 (W) mm^2^. A ground plane of the same dimensions as the substrate is placed on the bottom side of the substrate.

The RDRA array in the H-plane configuration is shown in Figure 1. The E-plane configuration is exactly the same with the single DRAs rotated by 90° counter clockwise around their axis. Both the arrays were simulated in Ansoft HFSS v19 [20] and later fabricated and measured. The measurement setup is shown in Figure 2. It consists of a 1 × 8 power divider (ZN8PD1-53+) (four ports terminated with matched loads); and the remaining four connected to a voltage-controlled attenuator (ZX73-2500+), voltage-controlled phase shifter (HMC928LP5E) and DRA element, as shown in Figure 2a. In addition, a low noise amplifier (ZX60-33LNC) feeds the power divider via the VNA. Figure 2b,c shows the feed network experimental setup and anechoic chamber measurement setup. The S-parameters of the four-element RDRA array in both configurations were measured by using a two-ports vector network analyzer (ZVB 20). For this purpose, two RDRAs were measured at a time; while the remaining two RDRAs were matched, terminated in the E- and H-plane array configurations. The same procedure was used for both element spacings.

The array factor (AF) for the array shown in Figure 1 can be written as [21]:AF = ∑| *w_n_* |e^jn(βdsinθs+*α*)^(1)
where *w_n_ = |w_n_| < α_n_* are the complex weights driving each antenna element, and *α_n_ = nα* is the inter-element phase shift required for the main beam scanning in the direction of θs. In addition, in Equation (1), four elements are considered, β is the wave number and a spherical coordinate system is adopted.
α = −βdsinθs(2)

This is the element-to-element phase shift required to steer the main beam in a direction θs. For example, for a uniformly excited array, the phase angles required to drive the four RDRAs of Figure 1 will be 1∠0, 1∠α, 1∠2α and 1∠3α, respectively, to point out the peak beam angle towards θs.

In the next section, the effects of mutual coupling on the gain and half-power beam width (HPBW) of the main beam scanning will be investigated for various scan angles.

Additionally, the electric (E) and magnetic (H) fields inside the DR element of the E- and H-plane configurations are presented in Figure 3 below. Figure 3a,b shows the E and H fields inside the DR in the H-plane configuration. It is clear from Figure 3a that the DR is resonating in its fundamental mode of TE^δ11^. Similarly, Figure 3c,d shows the E and H fields of the DR element in the E-plane configuration. It is clear from Figure 3c that the DR element is resonating in its fundamental mode of TE^1δ1^. The field distributions inside the DR elements in the array configuration remain the same.

## 3. Measured Results and Discussion

This section includes the measured results in terms of S-parameters, radiation pattern, gain and beam width plots.

### 3.1. S-Parameters

Figure 4 shows the fabricated DRA array. The measured reflection coefficient (|Sii|) plots for the E-plane and H-plane arrays for both inter-element spacings are shown in Figure 5 and Figure 6, respectively. The measured results show that both the E- and H-plane DRA arrays are resonating at 2.3 GHz for d = 0.5 λ, with an acceptable bandwidth of nearly 180 MHz. However, as the spacing reduces to 0.3 λ, the matching of the E-plane array deteriorates more than the H-plane array; this is evident from Figure 5b and Figure 6b. The impedance of the elements and hence, its matching, strongly depends upon the coupling; as the EM energy that is radiated by one element of the array becomes coupled to the nearby element of the array. This EM energy in the form of current reaches the terminal of the element, thereby modifying its impedance and matching; this is because impedance is the ratio of the voltage to current. The measured coupling coefficients (|Sij|) at the operating frequency of 2.3 GHz for both inter-element spacings are presented in Table 1 and Table 2 for the E- and H-plane configurations. Due to symmetry, only the coupling coefficients between the adjacent and far-away elements are shown. The tables reveal that as the inter-element spacing reduces the coupling between, the array element increases significantly for both configurations. Due to mutual coupling being a near-field phenomenon, as the spacing reduces, the electric (E) and magnetic (H) field becomes significantly stronger; thus, resulting in an increased coupling in both the E- and H-plane configurations. The coupling in the E-plane configuration is stronger than in the H-plane configuration for both inter-element spacings. This can be explained with the help of the H-field distribution inside the DR elements of both configurations, as shown in Figure 7. Due to the symmetry of both array configurations, the H-field distribution is shown only in the first two DR elements of the both the arrays. Figure 7a shows the H-field distribution inside the two DR elements of the H-plane array configuration. It can be clearly seen that the H fields in both the elements are opposite to each other, which results in the cancellation of the net H field; hence, this results in a low mutual coupling in the H-plane array configuration. Similarly, the H-field distribution inside the two DR elements in the E-plane array configuration is shown in Figure 7b. This figure reveals that the H field inside the DR elements is pointing in the same direction, resulting in the addition of the net H field. This results in an increased coupling in the E-field array configuration.

### 3.2. Radiation Patterns

Next, the effect of mutual coupling on the element radiation pattern is presented in Figure 8 and Figure 9 for the E- and H-plane configurations, respectively, for both inter-element spacings. The procedure discussed in [11] was adopted for the measurement of the active element patterns (AEPs). The measured radiation pattern of an isolated DRA is also shown in Figure 10 for comparison. By comparing Figure 8 and Figure 9 with Figure 10, it can be observed that the AEPs are deteriorated due to the mutual coupling effects. As can be noticed in Figure 8 and Figure 9, the AEPs of the H-plane configuration are much smoother than the AEPs of the E-plane configuration. This is because of the lower mutual coupling environment in the H plane than in the E plane. In order to show the effects of mutual coupling on the array patterns, the radiation patterns of a four-element RDRA H-plane array with inter-element spacing of 0.32 λ are presented in Figure 11a,b. The ideal (without coupling), simulated and measured patterns are shown for clarity. Figure 11a shows the pattern of the array when the main lobe/beam is pointing in the broadside direction. Figure 11b shows the pattern of the array when the main lobe is steered to 50°. It can be clearly observed that both patterns are significantly deteriorated when compared to the ideal patterns, due to the effects of mutual coupling.

### 3.3. Gain and Beam Width

Figure 12 and Figure 13 show the measured gain and beam width of the E- and H-plane array configurations. The gain of both the E- and H-plane array configurations are shown in Figure 12 for both inter-element spacings. From Figure 12a, it can be observed that for d = 0.5 λ, when the main beam of the array is in the broadside direction (θ = 0°), the gain of the E-plane array is 10.7 dBi; while that of the H-plane array is 9.8 dBi. However, as the beam is scanned off broadside towards the end fire direction, the gain of the E-plane array drops more sharply than the H-plane array. At θ = 90° (end fire direction), the gain of the E-plane array dropped to 6.4 dBi, while the gain of the H-plane dropped to 8.6 dBi. When the inter-element spacing is reduced to 0.32 λ, the gain of both array configurations drops due to an increase in mutual coupling; this is shown in Figure 12b.

For the E-plane array, the gain at θ = 0° is almost 9.2 dBi, which drops to 7.6 dBi when the main beam is scanned to θ = 42°. With further scanning, the gain improves and reaches 8.5i dB at θ = 90°. For the H-plane array, the gain is more stable. The gain at θ = 0° is 8.5 dBi and reaches 8.8 dBi when the main beam is pointing in θ = 90°. It can be concluded that in terms of gain performance, the H-plane array is outperforming the E-plane array. This can be explained by observing Figure 8 and Figure 9. As can be seen, the AEPs of the H plane are less affected than the AEPs of the E-plane array by mutual coupling. This results in a better performance of the H-plane array.

The performance comparison of the E- and H-plane arrays, in terms of beam width with main beam scanning, is presented in Figure 13. In Figure 13a, the beam width vs. beam scan angle for the inter-element spacing of d = 0.5 λ is presented for both array configurations. The beam width increase with scan angle is almost linear for both the E- and H-plane configurations. Nevertheless, in Figure 13b, the beam width is shown against the beam scan angle for d = 0.32 λ. In this case, the beam width of the E-plane array is increasing with the beam scan angle; starting from 39° at θ = 0° and reaching 78° at θ = 90°. For the H-plane array, the beam width at θ = 0° is 38°, which reaches the highest value of 62° at θ = 28°; and at θ = 90°, its value is 42°. The beam width is then almost constant when the main beam is scanned beyond 45°. The mutual coupling at d = 0.32 λ is severe in both the E- and H-plane array configurations; this results in an unusual performance in terms of beam width. However, the performance of the H-plane array is more stable than the E-plane array. This stability in the gain and beam width of the H-plane array configuration is attributed to the low mutual coupling in this configuration; this is because of the cancelation of the magnetic field (H) components in the array elements, as shown in Figure 7a. These results will provide excellent guidelines while designing a phased array based on dielectric resonators. Furthermore, various machine learning techniques can be utilized for the efficient correction and optimization of mutual coupling in order to recover the original array radiation pattern [22].

## 4. Conclusions

This paper presents a four-element RDRA array in the E- and H-plane configurations. The effects of mutual coupling on the radiation patterns, gain and beam width are studied for the inter-element spacing of 0.5 λ and 0.32 λ. The results indicate that the E-plane array is affected more severely than the H-plane array by mutual coupling. The gain of the E-plane array for each inter-element spacing drops more abruptly than the H-plane array. Similarly, the half-power beam width in the H-plane is more stable than the E-plane configuration. All these adverse effects are attributed to the increased mutual coupling environment in the E-plane array because the magnetic field (H) components in the array elements are in the same direction; thus, the net H field adds, resulting in an increased mutual coupling. Based on the presented results, it can be concluded that a DRA phased array in the H-plane configuration is more suitable than the E-plane array configuration.

## Figures and Tables

**Figure 1 sensors-22-06820-f001:**
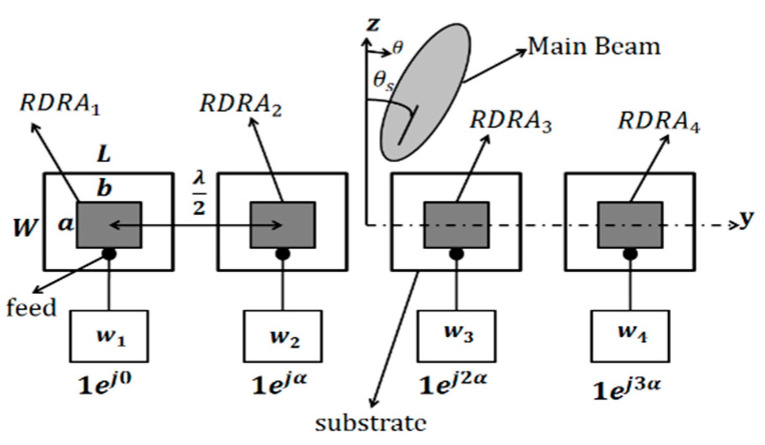
Schematic of the four-element scanning RDRA array.

**Figure 2 sensors-22-06820-f002:**
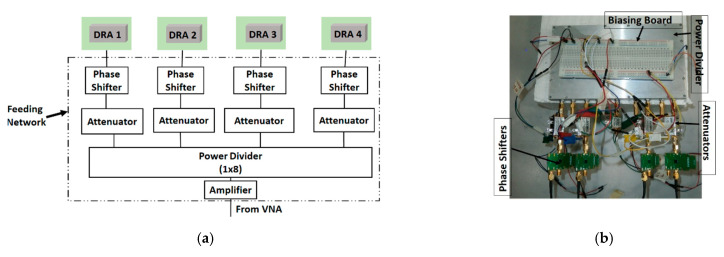
Measurement setup: (**a**) block diagram; (**b**) feed network; and (**c**) anechoic chamber setup.

**Figure 3 sensors-22-06820-f003:**
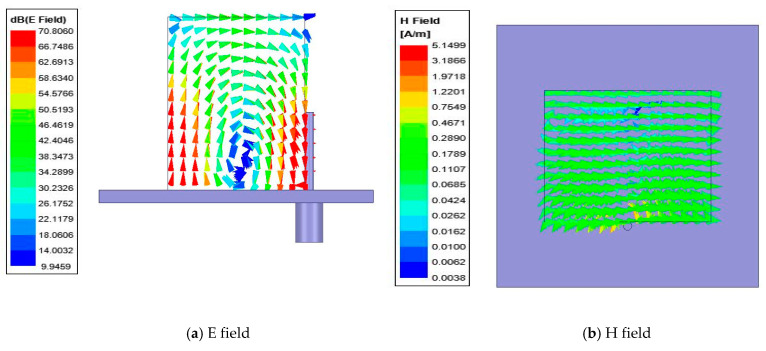
Field distribution inside the DR (**a**,**b**) H-plane configuration and (**c**,**d**) E-plane configuration.

**Figure 4 sensors-22-06820-f004:**
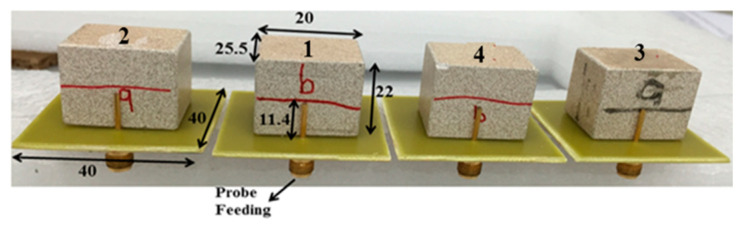
Prototype of the proposed DRA array.

**Figure 5 sensors-22-06820-f005:**
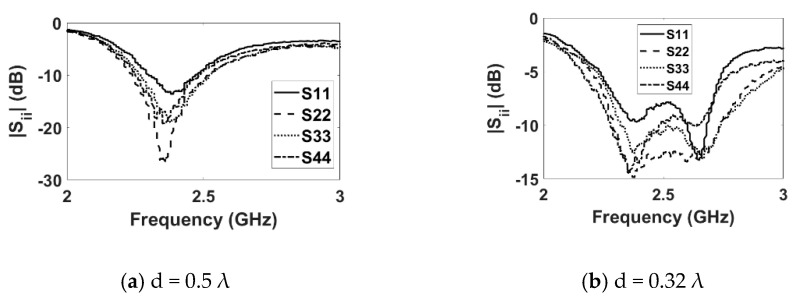
Measured reflection coefficient (E plane).

**Figure 6 sensors-22-06820-f006:**
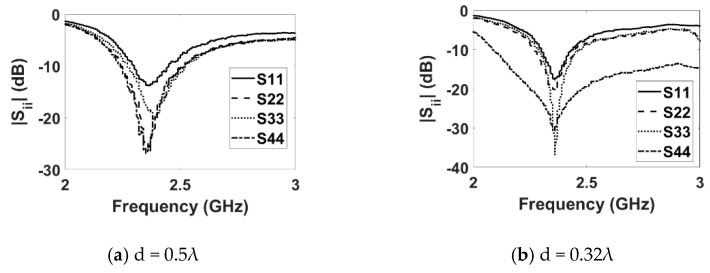
Measured reflection coefficient (H plane).

**Figure 7 sensors-22-06820-f007:**
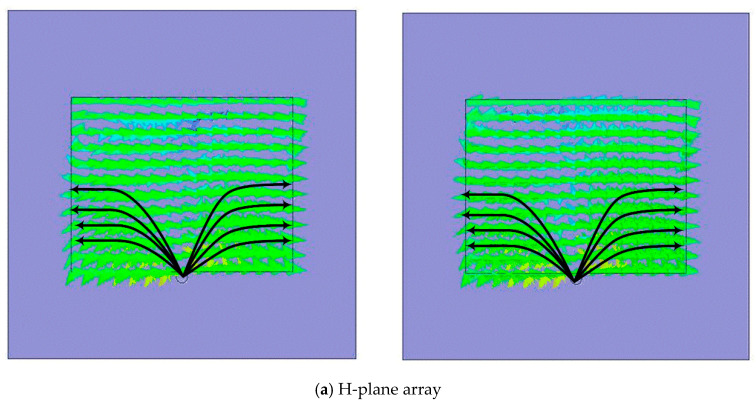
Magnetic field (H) distribution.

**Figure 8 sensors-22-06820-f008:**
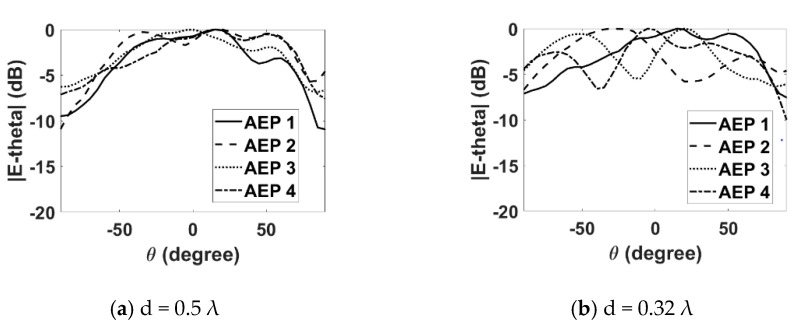
Measured active element patterns (E plane).

**Figure 9 sensors-22-06820-f009:**
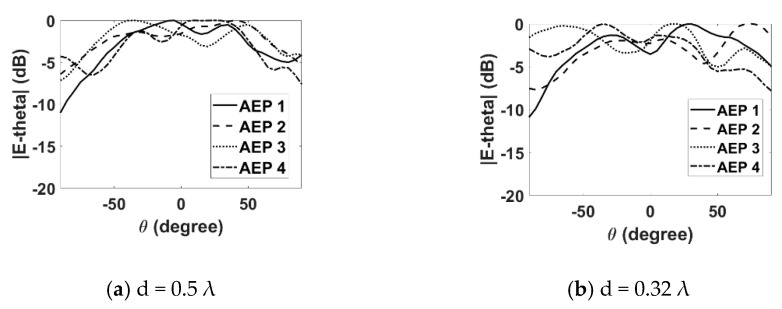
Measured active element patterns (H plane).

**Figure 10 sensors-22-06820-f010:**
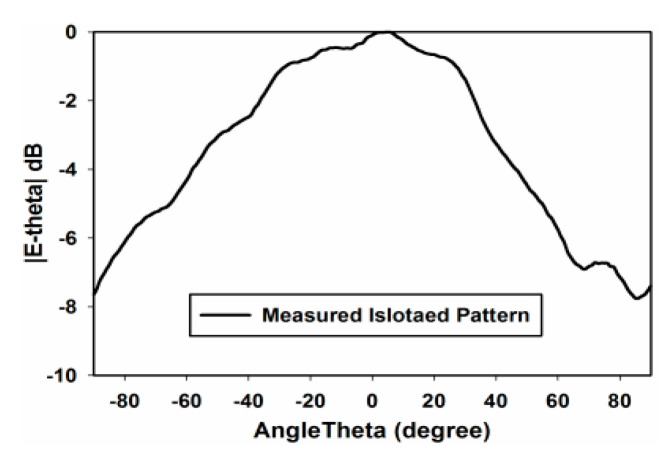
Measured isolated element pattern.

**Figure 11 sensors-22-06820-f011:**
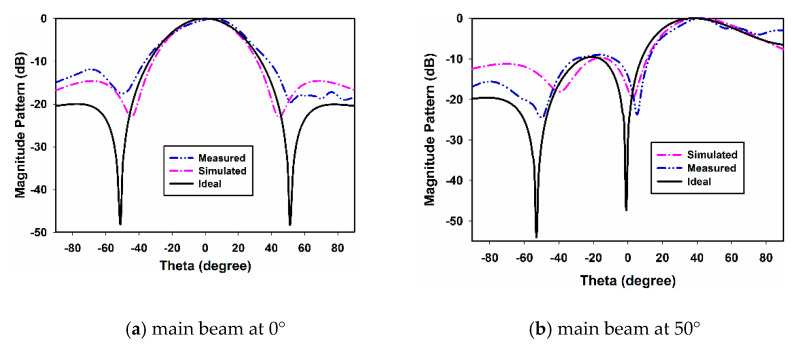
Array pattern.

**Figure 12 sensors-22-06820-f012:**
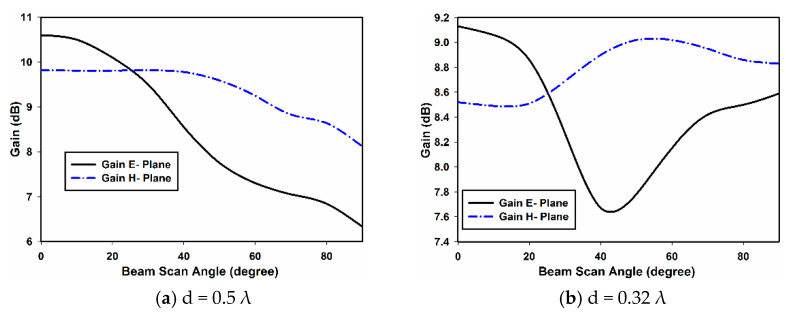
Measured gain of the E- and H-plane arrays.

**Figure 13 sensors-22-06820-f013:**
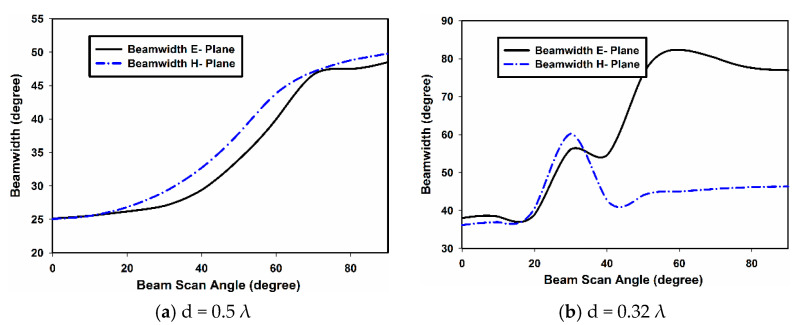
Measured beam width of the E- and H-plane arrays.

**Table 1 sensors-22-06820-t001:** Measured E-plane coupling coefficients (|Sij|) dB.

	S_12_	S_13_	S_14_	S_23_	S_24_
**d = 0.5 λ**	−28	−28.7	−28.6	−29.6	−28.7
**d = 0.32 λ**	−10.2	−16.9	−13.79	−25.4	−16.9

**Table 2 sensors-22-06820-t002:** Measured H-plane coupling coefficients (|Sij|) dB.

	S_12_	S_13_	S_14_	S_23_	S_24_
**d = 0.5 λ**	−17	−18	−17.2	−21	−18
**d = 0.32 λ**	−10.3	−11.3	−12.3	−11.4	−10.2

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
