# Peer review of "Effects of Mutual Coupling on Gain and Beam Width of a Linear Array of a Dielectric Resonator Antenna for Main Beam Scanning Applications"

_sensors, 2022, doi:10.3390/s22186820_

Round 1
Reviewer 1 Report
ï‚· Explain As inter-element spacing reduces the coupling between the array
element increases significantly for the fig 4 and fig 5.
ï‚· Need to explain why the coupling in the E-plane configuration is stronger
than the H-plane configuration for both fig 4 and 5.
ï‚· The inter-element spacing can be reduced further but why you reduced
the inter-element spacing to 0.32λ and need to mention the λ which is free
space wavelength or guided wavelength.
ï‚· If possible, draw Polar plot to substantiate the radiation pattern for the E
and H plane configuration.
ï‚· The author are suggested to plot E/H field distributions inside the
dielectric array.
ï‚· Need to identify and discuss the physical parameters those influence the
impedance.
Reviewer 2 Report
The authors presented global formulation of SST from a larger data base, which has practical applications. The updated global SST data will benefit future research by others. The main contribution is clearly explained. There are several minor factors the authors need to consider as below.
1. On line 8, SST should be explained.
2. For Figure 1, it is better to plot different lines with different shapes, patterns, textures, or labels on them for white-black friendly print
3. Table 1, λ is at another line.
4. Lots of measurements are provided. Can the authors tell more information why H-plane is more stable than E-plane in terms of gain and bandwidth?
Round 2
Reviewer 2 Report
The updated paper is good. In Fig. 8, there are some horizontal lines needed to be deleted, and you can still use different color curves with different sybmol to present the data.